# Investigating Different Context Types and Representations for Learning Word Embeddings

## Abstract

The number of word embedding models is growing every year. Most of them learn word embeddings based on the co-occurrence information of words and their contexts. However, it's still an open question what is the best definition of context. We provide the first systematical investigation of different context types and context representations for learning word embeddings. Comprehensive experiments are conducted to evaluate their effectiveness under 6 tasks, which give us some insights about context selection. We hope that this paper, along with the published code, can serve as a guideline of choosing context for our community.

## 1 Introduction

Recently, there is a growing research interest on word embedding models, where words are embedded into low-dimensional real vectors. Words that share similar meanings tend to have short distances in the vector space. The trained word embeddings are not only useful by themselves (e.g. used for calculating word similarities) but also effective when used as the input of the downstream models, such as part-of-speech tagging, chunking, named entity recognition (Collobert and Weston, 2008; Collobert et al., 2011) and text classification (Socher et al., 2013; Kim, 2014).

For almost all word embedding models, the training objectives are based on the Distributed Hypothesis (Harris, 1954), which can be stated as: "words that occur in the same contexts tend to have similar meanings". The "context" is usually defined as the words which precede and follow the target word within some fixed distance in most word embedding models with various architec-

tures (Bengio et al., 2003; Mnih and Hinton, 2007; Mikolov et al., 2013b; Pennington et al., 2014). Among them, Global Vectors (GloVe) proposed by Pennington et al. (2014), Continuous Skip-Gram (CSG) [1] and Continuous Bag-Of-Words (CBOW) proposed by Mikolov et al. (2013a) achieve state-of-the-art results on a wide range of linguistic tasks, and scales well to corpus with billion words.

Recently, Levy and Goldberg (2014b); Ling et al. (2015) [2] improve CSG and CBOW by introducing position-aware context representation, where each contextual word is associated with their relative position to the target word. Levy and Goldberg (2014a) propose DEPS, which takes the words that are connected to target word in dependency parse tree as context.

Despite all these efforts, there is still no clear answer to the following questions due to the lack of systematical comparison: 1) Is dependency-based context more reasonable than traditional linear one? 2) Do the relative position or the dependency relation between contextual word and target word contributes to the learning process? 3) Do different word embedding models have preferences for different contexts? 4) How different contexts affect models' performances on different tasks?

To answer these questions, we first classify word embedding models based on different context types (linear or dependency-based) and different context representations (word or bound word) in Table 1. We implement the models that previously not proposed and give systematical comparisons on a wide range of word similarity, word analogy, part-of-speech tagging, chunking, named entity recognition, and text classification dataset-

---

[1] Many researches refer Continuous Skip-Gram as SG. However, in order to distinguish linear (continuous) context and dependency-based context, we refer it as CSG.

[2] In these two papers, the description of position-aware context are quite different. However, their ideas is actually identical.

| Basic Model | Context Type / Context Representation | Linear | Dependency-based |
|---|---|---|---|
| generalized Skip-Gram | (unbound) word | CSG (Mikolov et al., 2013a) | this work |
| | bound word | Structured SG (Ling et al., 2015) POSIT (Levy and Goldberg, 2014b) | Deps (Levy and Goldberg, 2014a) |
| generalized Bag-Of-Words | (unbound) word | CBOW (Mikolov et al., 2013a) | this work |
| | bound word | CWINDOW (Ling et al., 2015) | this work |
| generalized GloVe | (unbound) word | GloVe (Pennington et al., 2014) | this work |
| | bound word | this work | this work |

Table 1: Research summarization of Generalized Skip-Gram, Bag-Of-Words and GloVe with different context types and context representations. For linear context, *bound word* indicates word associated with positional information. For dependency-based context, *bound word* indicates word associated with dependency relation.

s. Experimental results suggest that although it's hard to find any universal insight (i.e. one context works consistently better than the other), the characteristics of different contexts on different models are concluded according to specific tasks. We expect this paper to be a useful complement in the word embedding literature.

## 2 Methodology

In this section, we first introduce different contexts in detail and discuss their strength and weakness. We then show how CSG, CBOW and GloVe can be generalized to use these contexts.

### 2.1 Context Types

It is necessary to discover more effective ways of defining "context". In the current literature, there are mainly two types of contexts: linear (most word embedding models) and dependency-based (DEPS (Levy and Goldberg, 2014a)). Linear context is defined as the positional neighbours of the target word in texts. Dependency-based context is defined as the syntactic neighbours of the target word based on dependency parse tree, as shown in Figure 1 .

Compared to linear context, dependency-based context is more focused and can capture more long-range contexts. For example in Figure 1, linear context does not consider the word-context pair "discovers telescope", while dependency-based context contains this information. Dependency-based context can also exclude some uninformative word-context pairs like "with star" and "telescope with".

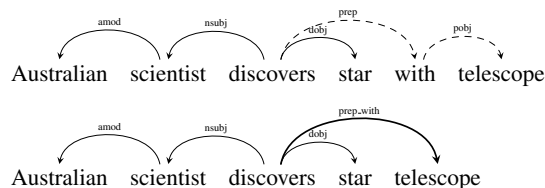

Figure 1: Illustration of dependency parse tree for sentence "Australian scientist discovers star with telescope". Note that preposition relation is collapsed in the bottom sub-figure, where *telescope* is considered as a direct modifier of *discovers*. We use the collapsed version of dependency in all of our experiments the same as Levy and Goldberg (2014b)

### 2.2 Context Representations

In CSG and CBOW, contexts are represented by words without any additional information. Levy and Goldberg (2014b); Ling et al. (2015) improve them by introducing position-bound words, where each contextual word is associated with their relative position to the target word. This allows C-SG and CBOW to distinguish different sequential positions and capture context's structural information. We name the method that bind additional information with the contextual word as bound (context) representation, as opposited to unbound (context) representation where word is used alone.

For dependency-based context, the original DEPS uses bound representation by default: words are associated with their dependency relation to the target word. Similar to bound representation in linear context type, this allows word embedding models to capture more dependency information. An example is shown in Table 2. In this paper, we

| Context Representation \ Context Type | Linear | Dependency-based |
|---|---|---|
| unbound | australian, scientist, star, with | scientist, star, telescope |
| bound | australian/-2, scientist/-1, star/+1, with/+2 | scientist/nsubj, star/dobj, telescope/prep_with |

Table 2: Illustration of bound and unbound representations under linear and dependency-based context types. This example is based on Figure 1 and the target word is "discovers".

|  | Linear (window size 1) | Dependency-based |
|---|---|---|
| $P$ | (**australian**, scientist) (**scientist**, australian) (**scientist**, discovers) (**discovers**, scientist) (**discovers**, star) … | (**australian**, scientist) (**scientist**, australian) (**scientist**, discovers) (**discovers**, scientist) (**discovers**, star) (**discovers**, telescope) … |
| $M$ | (**australian**, scientist) (**scientist**, australian, discovers) (**discovers**, scientist, star) … | (**australian**, scientist) (**scientist**, australian, discovers) (**discovers**, scientist, star, telescope) … |
| $\overline{M}$ | (**australian**, scientist, 1) (**scientist**, australian, 1) (**scientist**, discovers, 1) (**discovers**, scientist, 1) (**discovers**, star, 1) … | (**australian**, scientist, 1) (**scientist**, australian, 1) (**scientist**, discovers, 1) (**discovers**, scientist, 1) (**discovers**, star, 1) (**discovers**, telescope, 1) … |

Table 3: Illustration of collection $P$, $M$ and $\overline{M}$ for sentence "australian scientist discovers star with telescope". Unbound representation is used in this example. Words in the collections are **Bold**. Contexts and numbers in the collections are Normal.

also investigate the simpler context representation where no dependency relation is considered. This also makes a fair comparison with linear context models like CSG, CBOW and GloVe, since they do not use bound representation either.

Intuitively, bound representation should work better than unbound representation, since it is more sophisticated by considering position or dependency relation. However, this is not always the case in practice. The biggest drawback of word embeddings learned with bound context type is the ignorance of syntax. The bound representation already contains a certain degree of syntactic information, thus word embedding models can not learn it from the input word-context pairs. Another drawback is that bound context representation is sparse, especially for dependency-based context. There are 47 dependency relations in dependency parse tree. Although not every combination of dependency relations and words appear in the word-context pair collection, it still enlarges the context vocabulary about 5 times in practice.

Compared to context types (linear and dependency-based), the choice of context representations (bound and unbound) have more effects to the quality of the learned word embeddings. Bound representation transfers each contextual word into a new one, and the word-context pairs are changed completely. As for context types, a lot of word-context pairs in linear context type also appear in dependency-based context type. For example, in Table 2, "scientist" and "star" are considered as the contexts of "discovers" in both context types.

## 2.3 Generalization

For convenience, we first define the collection of word-context pairs as $P$. $P$ can be merged based on the words to form a collection $M$ with size of $|C|$. Each element $(w, c_1, c_2, .., c_{n_w}) \in M$ is the word $w$ and its contexts, where $n_w$ is the number of word $w$'s contexts. $P$ can also be merged based on both words and contexts to form a collection $\overline{M}$. Each element $(w, c, \#(w, c)) \in \overline{M}$ is the word $w$, context $c$, and the times they appear in collection $P$. An example of these collections is shown in Table 3.

### 2.3.1 Generalized Bag-Of-Words

The objective function of Generalized Bag-Of-Words (GBOW) is defined as:

$$\sum_{(w, c_1, .., c_{n_w}) \in M} \log p \left( w \middle| \sum_{i=1}^{n_w} \vec{c_i} \right) \qquad (1)$$

With negative sampling technique, the log probability is calculated by:

$$\log \sigma \left( \vec{w} \cdot \sum_{i=1}^{n_w} \vec{c_i} \right) - \sum_{k=1}^{K} \log \sigma \left( \vec{w_N} \cdot \sum_{1=i}^{n_w} \vec{c_i} \right) \qquad (2)$$

where $\sigma$ is the sigmoid function, $K$ is the negative sampling size, $\vec{w}$ and $\vec{c}$ is the vector for word $w$ and $c$ respectively. The negatively sampled word $w_N$ is randomly selected based on its unigram distribution $(\frac{\#(w)}{\sum_w \#(w)})^{ds}$, where $\#(w)$ is the number of times that word $w$ appears in the corpus, $ds$ is the distribution smoothing hyper-parameter which is usually defined as $0.75$.

Note that in the original CBOW (Mikolov et al., 2013a) with negative sampling technique, the probability is actually $p\left(c|\sum \vec{w}_i\right)$ instead of $p\left(w|\sum \vec{c}_i\right)$. In another word, the original CBOW uses the sum of word vectors to predict context. This works well for linear context. But for dependency-based context with bound representation, there is only one word available for predicting its context. For example in Figure 1, the context "scientist/nsubj" can only be predicted by word "discovers". However, a word can be predicted by the sum of several contexts. Due to this reason, we exchange the role of word and context in GBOW. The negative sampling objective is also changed from context $c_N$ to word $w_N$.

### 2.3.2 Generalized Skip-Gram

For generalized Skip-Gram (GSG), the definition is straightforward and the objective function actually needs no modification (Levy and Goldberg, 2014b). However, in order to make it consistent with our GBOW, we also exchange the role of word and context. The objective function of GSG is defined as:

$$\sum_{(w,c)\in P} \log p\left(w|\vec{c}\right)$$

$$= \sum_{(w,c)\in P} \left[ \log \sigma\left(\vec{w}\cdot\vec{c}\right) - \sum_{k=1}^{K} \log \sigma\left(\vec{w_N}\cdot\vec{c}\right) \right] \tag{3}$$

### 2.3.3 GloVe

Unlike GSG and GBOW, GloVe explicitly optimizes a log-bilinear regression model based on word co-occurrence matrix. Since GloVe is already a very generalized model, with the previous defined collection $\overline{M}$, the final objective function is written as:

$$\sum_{(w,c)\in\overline{M}} f(\#(w,c))(\vec{w}\cdot\vec{c}+\vec{b_w}+\vec{b_c}-\log\#(w,c)) \tag{4}$$

where $\vec{b_w}$ and $\vec{b_c}$ are biases for word and context. $f$ is a non-decreasing weighting function and ensures that large $\#(w,c)$ is not over-weighted.

Note that the inputs of GSG, GBOW and Glove are the collections $P$, $M$ and $\overline{M}$ respectively. Once the corpus and hyper-parameters are fixed, these collections (and thus the learned word embeddings) are determined only by the choice of context types and representations.

## 3 Experiments

We evaluate the effectiveness of different context types and representations on word similarity, word analogy, part-of-speech tagging, chunking, named entity recognition, and text classification tasks. In this section, we first describe the training details of word embedding models. We then report and discuss the experimental results on each task. Detailed numerical results can be found in Supplemental Material.

### 3.1 Training Details

Previously, the `word2vecf` toolkit [3] (Levy et al., 2015) extends the `word2vec` toolkit [4] (Mikolov et al., 2013b) to accept the input of collection $P$ rather than raw corpus. This makes CSG model accept arbitrary contexts (e.g. dependency-based context). However, CBOW and GloVe are not considered in that toolkit. We implement `word2vecPM` toolkit, a further extension of `word2vecf`, which supports generalized SG, CBOW and GloVe with the input of collection $P$, $M$ and $\overline{M}$ respectively.

We use English Wikipedia (August 2013 dump) as the training corpus in all of our experiments. The Stanford CoreNLP (Manning et al., 2014) is used for dependency parsing. All words and contexts are converted to lower case after parsing. Words and contexts that appear less than 100 times in the collection $P$ are directly ignored. Note that this is slightly different from ignoring rare words that appear less than 100 times in the vocabulary based on the corpus, since each word may appear more times in the collection than that in the vocabulary.

Most hyper-parameters are the same as Levy et al. (2015)'s best configuration. For example, negative sampling size $K$ is set to 5 for GSG and 2 for GBOW. Distribution smoothing $cds$ is set to 0.75. No dynamic context or "dirty" sub-sampling is used. The window size $wn$ is fixed to 2 for constructing linear context, which ensures the number of the (merged) word-context pair collection for both linear context and dependency-based context is comparable. The number of iteration is set to 2, 5 and 30 for GSG, GBOW and GloVe respectively. Unless otherwise noted, the number of word embedding dimension is set to 500. Since the aim

---

[3] https://bitbucket.org/yoavgo/word2vecf
[4] http://code.google.com/p/word2vec/

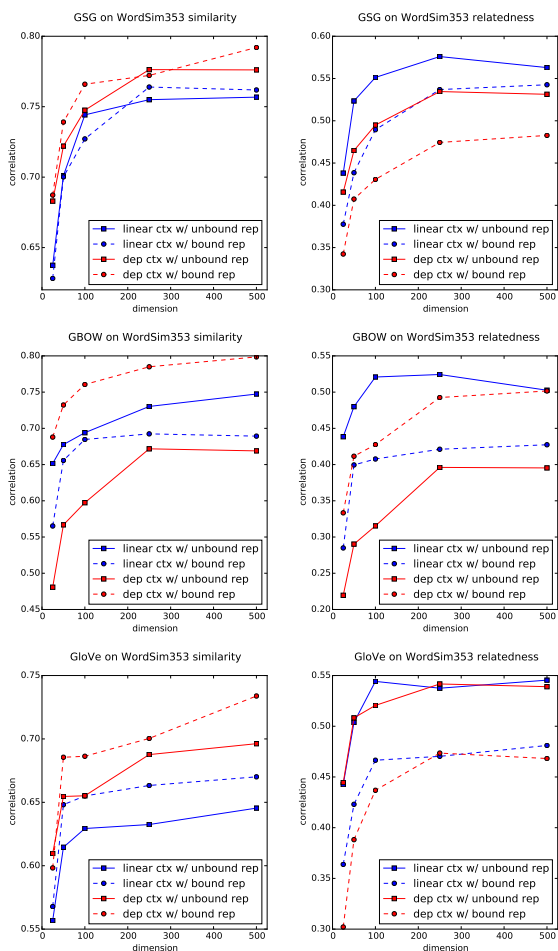

Figure 2: Results on WordSim353 (similarity and relatedness) dataset.

of this paper is not comparing the performance of different word embedding models, the results of GSG, GBOW and GloVe are reported respectively.

### 3.2 Word Similarity Task

Word similarity task aims at producing semantic similarity scores of word pairs, which are compared with the human scores using Spearman's correlation. The cosine distance is used for generating similarity scores between two word vectors. WordSim353 (Finkelstein et al., 2001) dataset with similarity and relatedness partition (Zesch et al., 2008; Agirre et al., 2009) is used for this task.

Previous researches (Levy and Goldberg, 2014a; Melamud et al., 2016) conclude that compared to linear context, dependency-based context can capture more functional similarity (e.g. tiger/cat) rather than topical similarity (relatedness) (e.g., tiger/jungle). However, their experiments do not distinguish the effect of differen-

t context representations: unbound representation is used for linear context (Mikolov et al., 2013b) while bound representation is used for dependency-based context (Levy and Goldberg, 2014a). Moreover, only CSG model is compared.

We revisit those claims based on more systematical experiments. As shown in Figure 2's top-left sub-figure, compared to linear context (solid and dotted blue line), the better results of dependency-based context for GSG and GloVe (solid and dotted red line) on ws353's similarity partition confirms its ability of capturing functional similarity. However, the good performances of dependency-based context do not fully transfer to GBOW. Although dependency-based context with bound representation (dotted red line) for GBOW is still the best performer, dependency-based context with unbound representation (solid red line) for GBOW performs worst on ws353's similarity partition. Distinguishing bound representation from unbound representation is important.

Note that the results are also reversed on ws353's relatedness partition (Figure 2's right sub-figures), which shows the use of linear context is more suitable for capturing topical similarity.

Overall, dependency-based context type does not get all the credit for capturing functional similarity. Context representations play an important role for word similarity task. It's only safe to say that dependency-based context captures functional similarity with the "help" of bound representation. In contrast, linear context type captures topical similarity with the "help" of unbound representation.

### 3.3 Word Analogy Task

Word analogy task aims at answering the questions like "a is to b as c is to __ ?". For example, "London is to Britain as Tokyo is to Japan". We follow the evaluation protocol in Levy and Goldberg (2014b), answering the questions using both 3CosAdd (additive) and 3CosMul (multiplicative) functions. Our experiments show that 3CosMul works consistently better than 3CosAdd, thus only the results of 3CosMul are reported. We follow previous researches and use Google's analogy dataset (Mikolov et al., 2013a) (with semantic and syntactic partition) in our experiments.

As shown in Figure 3, we observe that context representation plays an important role in word analogy task. The choice of context representa-

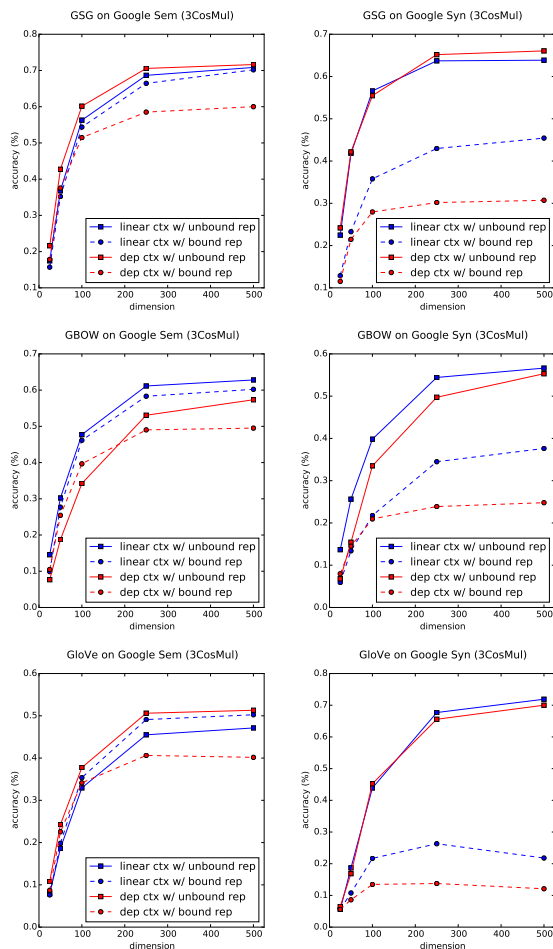

Figure 3: Results on Google (Sem and Syn) dataset.

tion (word or bound word) actually has much larger impact than the choice of context type (linear or dependency). The results on Google Syn dataset (Figure 3's sub-figures in the second column) is perhaps the most evident. The performance of linear context and dependency-based context with unbound representation is similar. However, when bound representation is used, the performance of GSG and GBOW drops more than 30 percent for dependency-based context and around 20 percent for linear context. The main reason for this phenomenon is that the bound representation already contains syntactic information, thus word embedding models can not learn it from the input word-context pairs. It can also be observed that GloVe is more sensitive to different context representations than Skip-Gram and CBOW, which is probably due to its explicitly defined/optimized objective function.

### 3.4 POS, Chunking and NER Tasks

Although intrinsic evaluations like word similarity and word analogy tasks could provide direct

insights of different context types and representations, the experimental results above cannot be directly translated to the typical uses of word embeddings. For example, these tasks aren't necessarily correlated with downstream tasks' performances, as shown in (Schnabel et al., 2015; Linzen, 2016; Chiu et al., 2016). More extrinsic tasks should be considered.

In this subsection, we evaluate the effectiveness of different word embedding models with different contexts on Part-of-Speech Tagging (POS), Chunking and Named Entity Recognition (NER) tasks. These tasks can be categorized as sequence labeling. It aims at automatically assigning words in texts with labels. CoNLL 2000 shared task [5] is used as benchmark for POS and Chunking. CoNLL 2003 shared task [6] is used as benchmark for NER.

Inspired by the evaluation protocol used in Kiros et al. (2015), we restrict the predicting model to simple linear classifier. The classifier's input for predicting the label of word $w_i$ is simply the concatenation of vectors $\vec{w_{i-2}}, \vec{w_{i-1}}, \vec{w_i}, \vec{w_{i+1}}, \vec{w_{i+2}}$. This ensures the quality of embedding models is directly evaluated, and their strengths and weaknesses are easily observed.

As shown in Figure 4, the overall trends of GSG, GBOW and GloVe are similar. When the same context type is used, bound representation (dotted line) outperforms unbound representation (solid line) on all datasets. Sequence labeling tasks tend to classify words with the same syntax to the same category. The ignorance of syntax for word embeddings which are learned by bound representation becomes beneficial. Moreover, dependency-based context type works better than linear context type in most cases. These results suggest that linear context type with unbound representations (as in traditional CSG and CBOW) may not be the best choice of input word vectors for sequence labeling. Bound representations should always be used and dependency-based context type is also worth considering. Again, similar to that on word analogy task, GloVe is more sensitive to different context representations than Skip-Gram and CBOW on sequence labeling tasks.

---

[5] http://www.cnts.ua.ac.be/conll2000/chunking
[6] http://www.cnts.ua.ac.be/conll2003/ner

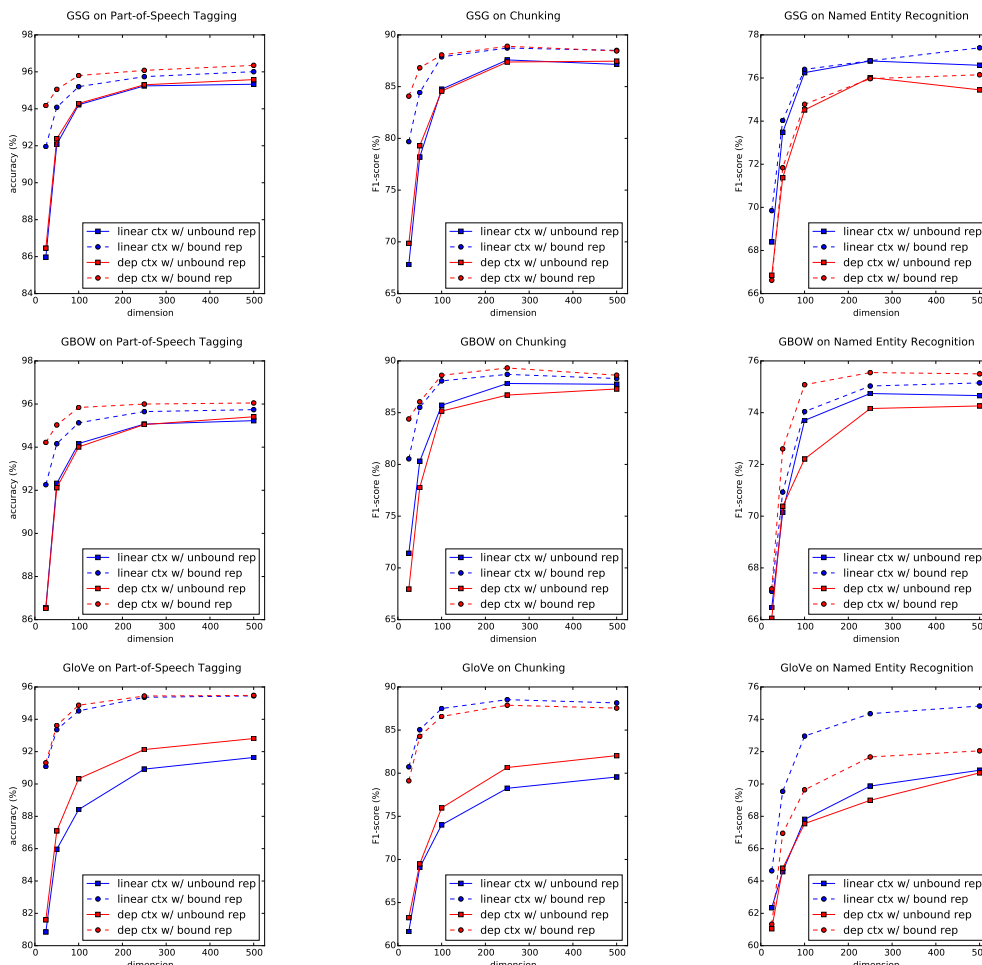

Figure 4: Results on POS, Chunking and NER tasks.

## 3.5 Text Classification Task

Finally, we evaluate the effectiveness of different word embedding models with different contexts on text classification task. Text classification is one of the most popular and well-studied tasks in natural language processing. Recently, deep neural networks are dominant on this task (Socher et al., 2013; Kim, 2014; Dai and Le, 2015). They often need pre-trained word embeddings as inputs to improve their performances. Similar to the previous evaluation of sequence labeling tasks, instead of building complex deep neural networks, we use a simpler classification method called Neural Bag-of-Words to directly evaluate the word embeddings: texts are first represented by the sum of their belonging words' vectors, then a Logistic Regression Classifier is built upon them for classification.

Different word embedding models are evaluated on 5 text classification datasets. The first 3 datasets are sentence-level: short movie review sentiment (MR) (Pang and Lee, 2005), customer product re-

| Model | Context Type | Context Rep. | Sentence-level | | | Document-level | |
|---|---|---|---|---|---|---|---|
| | | | MR | CR | Subj | RT-2k | IMDB |
| GSG | linear | word | 76.1 | 78.3 | 90.9 | 83.5 | 85.2 |
| | | bound | 75.3 | 79.0 | 90.4 | 82.2 | 85.2 |
| | dep | word | 76.0 | 77.7 | 90.7 | 84.8 | 85.1 |
| | | bound | 75.0 | 77.5 | 90.0 | 84.7 | 84.5 |
| GBOW | linear | word | 74.9 | 77.9 | 90.4 | 82.0 | 85.0 |
| | | bound | 74.1 | 77.8 | 90.3 | 80.7 | 84.1 |
| | dep | word | 75.0 | 77.6 | 90.1 | 82.4 | 84.9 |
| | | bound | 73.5 | 78.2 | 89.9 | 80.7 | 83.4 |
| GloVe | linear | word | 73.4 | 76.7 | 89.6 | 79.2 | 83.5 |
| | | bound | 73.2 | 77.5 | 90.0 | 79.8 | 83.4 |
| | dep | word | 74.0 | 77.7 | 89.5 | 81.3 | 83.5 |
| | | bound | 72.5 | 76.7 | 88.8 | 79.2 | 83.5 |
| Random word embeddings | | | 63.9 | 72.8 | 79.9 | 72.2 | 77.2 |

Table 4: Results on 5 text classification datasets.

views (CR) (Nakagawa et al., 2010), and subjectivity/objectivity classification (SUBJ) (Pang and Lee, 2004). The other 2 datasets are document-level with multiple sentences: full-length movie review (RT-2k) (Pang and Lee, 2004), and IMDB movie review (IMDB) (Maas et al., 2011).

As shown in Table 4, pre-trained word embeddings outperform random word embeddings by a large margin. This further strengthens previous

researches that pre-trained word embeddings are crucial for text classification. Unlike that on previous tasks, different models' results are actually very similar on text classification task. Text classification has less focus on syntax and function similarity. This leads to the phenomenon that models which use bound representation perform worse than those which use unbound representation on all datasets except CR. Models that use dependency-based context type and linear context type are comparable. These observations suggest that simple linear context type with unbound representations (as in traditional CSG and CBOW) is still the best choice of pre-training word embeddings for text classification, which is already used in most researches.

## 4  Related Work

Previously, there are researches which directly compare different word embedding models. Lai et al. (2016) compare 6 word embedding models using different corpora and hyper-parameters. Levy and Goldberg (2014c) show the theoretical equivalence of CSG and PPMI matrix factorization. Levy et al. (2015) further discuss the connections between 4 word embedding models (PPMI, PPMI+SVD, CSG, GloVe) and re-evaluate them with the same hyper-parameters. Suzuki and Nagata (2015) investigate different configurations of CSG and Glove, then merge them into a unified form. Yin and Schutze (2016) propose 4 ensemble methods and show their effectiveness over individual word embeddings.

There are also researches which focus on evaluating different context types in learning word embeddings. Vulic and Korhonen (2016) compare CSG and dependency-based models on various languages. The results suggest that dependency-based models are able to detect functional similarity in English. However, the advantages of dependency-based context over linear context on other languages are not as promising as that on English. Bansal et al. (2014) investigate different embedding models for parsing task and show that dependency-based context is more suitable than linear context on this task. Melamud et al. (2016) investigate the performance of CSG, Deps and a substitute-based word embedding models (Yatbaz et al., 2012) [7], which shows that different types of

---

[7]We do not consider this type of context, since it performs consistently worse than the other two context types. The

intrinsic tasks have clear preference to particular types of contexts. On the other hand, for extrinsic tasks, the optimal context types need to be carefully tuned on specific dataset. However, context representations (bound and unbound) are not evaluated in these models. Moreover, they focus only on the more popular and intuitive CSG model, but not on CBOW and GloVe.

## 5  Conclusion

To the best of our knowledge, this paper provides the first systematical investigation of different context types and representations for learning word embeddings. We evaluate different models on intrinsic property analysis (word similarity and word analogy), sequence labeling tasks (POS, Chunking and NER) and text classification task.

Overall, the tendency of different models on different tasks is similar. However, most tasks have clear preference for different context types and representations. Context representations play a more important role than context types for learning word embeddings. More precisely: 1) Unbound representation is more suitable for syntactic word analogy than bound representation. Bound representation already contains syntactic information, which makes it difficult to learn syntactic aware word embeddings based on the input word-context pairs. 2) No matter which type of context to be used, bound representation is essential for sequence labeling tasks, which benefits from its ability of capturing functional similarity. In contrast, unbound representation, which is suitable for capturing topical similarity, doesn't contribute to sequence labeling tasks. 3) Linear context with unbound representation (Skip-Gram) is still the best choice for text classification task. Linear context is enough for capturing topical similarity compared to dependency-based context. Words' position information is generally useless for text classification, which makes bound representation contribute less to this task.

In the spirit of transparent and reproducible experiments, the `word2vecPM` toolkit in Supplemental Material will be published online. We hope researchers will take advantage of the code for further improvements and applications to other tasks.

---

same observation is made by Melamud et al. (2016); Vulic and Korhonen (2016)

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

## A Supplemental Material

### A.1 Numerical Results

For simplicity and clarity, most experimental results are shown in the form of line chart. However, numerical results are more accurate and can be directly used by other researches. We list them in Table 5, 6 and 7. Upon acceptance, they will be added to the final version.

| Model | Context Type | Context Rep. | WS353 Sim | Related |
|-------|-------------|-------------|------|---------|
| GSG | linear | word | .757 | **.563** |
| | | bound | .762 | .543 |
| | dep | word | .776 | .531 |
| | | bound | **.792** | .483 |
| GBOW | linear | word | .747 | **.503** |
| | | bound | .689 | .427 |
| | dep | word | .669 | .395 |
| | | bound | **.799** | .502 |
| GloVe | linear | word | .645 | **.545** |
| | | bound | .670 | .481 |
| | dep | word | .696 | .539 |
| | | bound | **.734** | .468 |

Table 5: Numerical results on word similarity task. Best results in group are marked **Bold**.

| Model | Context Type | Context Rep. | Google Sem | Syn |
|-------|-------------|-------------|------|-----|
| GSG | linear | word | .708 | .639 |
| | | bound | .702 | .454 |
| | dep | word | **.716** | **.661** |
| | | bound | .600 | .307 |
| GBOW | linear | word | **.628** | **.566** |
| | | bound | .602 | .376 |
| | dep | word | .573 | .553 |
| | | bound | .495 | .248 |
| GloVe | linear | word | .471 | **.719** |
| | | bound | .502 | .218 |
| | dep | word | **.513** | .700 |
| | | bound | .402 | .121 |

Table 6: Numerical results on word analogy task. Best results in group are marked **Bold**.

| Model | Context Type | Context Rep. | POS | Chunking | NER |
|-------|-------------|-------------|------|----------|------|
| GSG | linear | word | 95.3 | 87.2 | 76.6 |
| | | bound | 96.0 | **88.5** | **77.4** |
| | dep | word | 95.6 | 87.5 | 75.5 |
| | | bound | **96.3** | **88.5** | 76.2 |
| GBOW | linear | word | 95.2 | 87.7 | 74.7 |
| | | bound | 95.7 | 88.3 | 75.2 |
| | dep | word | 95.4 | 87.3 | 74.3 |
| | | bound | **96.0** | **88.6** | **75.5** |
| GloVe | linear | word | 91.6 | 79.6 | 70.8 |
| | | bound | **95.5** | **88.2** | **74.8** |
| | dep | word | 92.8 | 82.0 | 70.7 |
| | | bound | **95.5** | 87.5 | 72.0 |

Table 7: Numerical results on POS, Chunking and NER tasks. Best results in group are marked **Bold**.

