# Peer review of "Investigating Different Context Types and Representations for Learning Word Embeddings"

_ACL 2017 — decision unknown_

[Official Review · Reviewer 1 · rating 2 · confidence 4]
soundness 3 · originality 3 · clarity 3 · impact 3 · substance 3 · appropriateness 3 · meaningful comparison 3 · presentation format Poster

- Strengths:

This paper presents a 2 x 2 x 3 x 10 array of accuracy results based on
systematically changing the parameters of embeddings models:

(context type, position sensitive, embedding model, task), accuracy

- context type ∈ {Linear, Syntactic}
- position sensitive ∈ {True, False}
- embedding model ∈ {Skip Gram, BOW, GLOVE}
- task ∈ {Word Similarity, Analogies, POS, NER, Chunking, 5 text classific.
tasks}

The aim of these experiments was to investigate the variation in
performance as these parameters are changed. The goal of the study itself
is interesting for the ACL community and similar papers have appeared
before as workshop papers and have been well cited, such as Nayak et al.'s
paper mentioned below.

- Weaknesses:
Since this paper essentially presents the effect of systematically changing the

context types and position sensitivity, I will focus on the execution of the
investigation and the analysis of the results, which I am afraid is not 
satisfactory.

A) The lack of hyper-parameter tuning is worrisome. E.g.
   - 395 Unless otherwise notes, the number of word embedding dimension is set
to 500.
   - 232 It still enlarges the context vocabulary about 5 times in practice.
   - 385 Most hyper-parameters are the same as Levy et al' best configuration.

  This is worrisome because lack of hyperparameter tuning makes it difficult to
make statements like method A is better than method B. E.g. bound methods may
perform better with a lower dimensionality than unbound models, since their
effective context vocabulary size is larger.

B) The paper sometimes presents strange explanations for its results. E.g.
   - 115 "Experimental results suggest that although it's hard to find any 
universal insight, the characteristics of different contexts on different
models are concluded according to specific tasks."

   What does this sentence even mean? 

   - 580 Sequence labeling tasks tend to classify words with the same syntax 
to the same category. The ignorance of syntax for word embeddings which  are
learned by bound representation becomes beneficial. 

   These two sentences are contradictory, if a sequence labeling task
   classified words with "same syntax" to same category then syntx becomes
   a ver valuable feature. Bound representation's ignorance of syntax
   should cause a drop in performance just like other tasks which does not
   happen.

C) It is not enough to merely mention Lai et. al. 2016 who have also done a
   systematic study of the word embeddings, and similarly the paper 
   "Evaluating Word Embeddings Using a Representative Suite of Practical
   Tasks", Nayak, Angeli, Manning. appeared at the repeval workshop at 
   ACL 2016. should have been cited. I understand that the focus of Nayak
   et al's paper is not exactly the same as this paper, however they
   provide recommendations about hyperparameter tuning and experiment
   design and even provide a web interface for automatically running
   tagging experiments using neural networks instead of the "simple linear
   classifiers" used in the current paper.

D) The paper uses a neural BOW words classifier for the text classification
tasks
   but a simple linear classifier for the sequence labeling tasks. What is
   the justification for this choice of classifiers? Why not use a simple
   neural classifier for the tagging tasks as well? I raise this point,
   since the tagging task seems to be the only task where bound
   representations are consistently beating the unbound representations,
   which makes this task the odd one out. 

- General Discussion:
Finally, I will make one speculative suggestion to the authors regarding
the analysis of the data. As I said earlier, this paper's main contribution is
an
analysis of the following table.
(context type, position sensitive, embedding model, task, accuracy)
So essentially there are 120 accuracy values that we want to explain in
terms of the aspects of the model. It may be beneficial to perform
factor analysis or some other pattern mining technique on this 120 sample data.

[Official Review · Reviewer 2 · rating 4 · confidence 4]
soundness 3 · originality 3 · clarity 5 · impact 3 · substance 5 · appropriateness 5 · meaningful comparison 3 · presentation format Oral Presentation

- Strengths: 
Evaluating bag of words and "bound" contexts from either dependencies or
sentence ordering is important, and will be a useful reference to the
community. The experiments were relatively thorough (though some choices could
use further justification), and the authors used downstream tasks instead of
just intrinsic evaluations.

- Weaknesses: 
The authors change the objective function of GBOW from p(c|\sum w_i) to
p(w|\sum c_i). This is somewhat justified as dependency-based context with a
bound representation only has one word available for predicting the context,
but it's unclear exactly why that is the case and deserves more discussion.
Presumably the
non-dependency context with a bound representation would also suffer from this
drawback? If so, how did Ling et al., 2015 do it? Unfortunately, the authors
don't compare any results against the original objective, which is a definite
weakness. In addition, the authors change GSG to match GBOW, again without
comparing to the original objective. Adding results from word vectors trained
using the original GBOW and GSG objective functions would justify these changes
(assuming the results don't show large changes).
The hyperparameter settings should be discussed further. This played a large
role in Levy et al. (2015), so you should consider trying different
hyperparameter values. These depend pretty heavily on the task, so simply
taking good values from another task may not work well.

In addition, the authors are unclear on exactly what model is trained in
section 3.4. They say only that it is a "simple linear classifier". In section
3.5, they use logistic regression with the average of the word vectors as
input, but call it  a Neural Bag-of-Words model. Technically previous work also
used this name, but I find it misleading, since it's just logistic regression
(and hence a linear model, which is not something I would call "Neural"). It is
important to know if the model trained in section 3.4 is the same as the model
trained in 3.5, so we know if the different conclusions are the result of the
task or the model changing. 

- General Discussion: This paper evaluates context taken from dependency parses
vs context taken from word position in a given sentence, and bag-of-words vs
tokens with relative position indicators. This paper is useful to the
community, as they show when and where researchers should use word vectors
trained using these different decisions. 

- Emphasis to improve:
The main takeaway from this paper that future researchers will use is given at
the end of 3.4 and 3.5, but really should be summarized at the start of the
paper. Specifically, the authors should put in the abstract that for POS,
chunking, and NER, bound representations outperform bag-of-words
representations, and that dependency contexts work better than linear contexts
in most cases. In addition, for a simple text classification model, bound
representations perform worse than bag-of-words representations, and there
seemed to be no major difference between the different models or context types.

- Small points of improvement: 
Should call "unbounded" context "bag of words". This may lead to some confusion
as one of the techniques you use is Generalized Bag-Of-Words, but this can be
clarified easily.
043: it's the "distributional hypothesis", not the "Distributed Hypothesis". 
069: citations should have a comma instead of semicolon separating them.
074: "DEPS" should be capitalized consistently throughout the paper (usually it
appears as "Deps"). Also should be introduced as something like dependency
parse tree context (Deps).
085: typo: "How different contexts affect model's performances..." Should have
the word "do".

[Official Review · Reviewer 3 · rating 4 · confidence 5]
soundness 3 · originality 3 · clarity 5 · impact 3 · substance 4 · appropriateness 5 · meaningful comparison 3 · presentation format Poster

- Strengths:

This paper systematically investigated how context types (linear vs
dependency-based) and representations (bound word vs unbound word) affect word
embedding learning. They experimented with three models (Generalized
Bag-Of-Words, Generalized Skip-Gram and Glove) in multiple different tasks
(word similarity, word analogy, sequence labeling and text classification).
Overall, 
1)            It is well-written and structured.
2)            The experiments are very thoroughly evaluated. The analysis could
help
researchers to choose different word embeddings or might even motivate new
models. 
3)            The attached software can also benefit the community. 

- Weaknesses:

 The novelty is limited. 

- General Discussion:

For the dependency-based context types, how does the dependency parsing affect
the overall performance? Is it fair to compare those two different context
types since the dependency-based one has to rely on the predicted dependency
parsing results (in this case CoreNLP) while the linear one does not?